# Association between Spinal Cord Injury and Alcohol Dependence: A Population-Based Retrospective Cohort Study

**DOI:** 10.3390/jpm12030473

**Published:** 2022-03-16

**Authors:** Ching-Hui Chuang, Po-Cheng Chen, Chyi-Huey Bai, Yi-Lin Wu, Ming-Chao Tsai, Chieh-Yu Li

**Affiliations:** 1Department of Nursing, Meiho University, Pingtung 912, Taiwan; helen.ch.chuang@gmail.com; 2Department of Rehabilitation, Kaohsiung Chang Gung Memorial Hospital, Kaohsiung 833, Taiwan; b9302081@cgmh.org.tw; 3School of Public Health, Taipei Medical University, Taipei 110, Taiwan; baich@tmu.edu.tw; 4Department of Nursing, College of Medicine, National Cheng Kung University, Tainan 704, Taiwan; lynnewu727@gmail.com; 5Department of Internal Medicine, Division of Hepato-Gastroenterology, Kaohsiung Chang Gung Memorial Hospital and Chang Gung University College of Medicine, Kaohsiung 833, Taiwan; 6Graduate Institute of Clinical Medical Sciences, College of Medicine, Chang Gung University, Taoyuan 333, Taiwan; 7Department of Nursing, Kaohsiung Chang Gung Memorial Hospital, Kaohsiung 833, Taiwan; chingju11@cgmh.org.tw

**Keywords:** spinal cord injury, alcohol dependence, National Health Insurance Research Database

## Abstract

Spinal cord injury (SCI) is a devastating disorder. Alcohol abuse has been recognized as hindering SCI patients from rehabilitation, thus leading to longer length of days and poorer prognosis. This article aimed to investigate the association between spinal cord injury (SCI) and alcohol dependence. Data were derived from the National Health Insurance Research Database (NHIRD). The incidence of alcohol dependence between SCI and non-SCI groups was compared. Other possible risk factors were also analyzed. Patients (N = 5670) with SCI from 2000 to 2009 were initially assessed for eligibility. After propensity score matching, 5639 first-time SCI survivors were included. The Cox proportional hazard regression model was used to assess differences in the incidence of alcohol dependence syndrome. Based on the adjusted hazard ratios (HR), the SCI group had a higher hazard for alcohol dependence syndrome compared to the non-SCI group (adjusted HR: 1.39, 95% CI: 1.03~1.86, *p* = 0.0305). The injury level did not have an impact on the incidence of alcohol dependence syndrome. A higher incidence of alcohol dependence syndrome was related to male patients, lower insurance levels, higher Deyo’s CCI, and psychiatric OPD times. A lower incidence of alcohol dependence syndrome was related to elder age. The incidence of alcohol dependence increased after the occurrence of SCI and was also related to age, sex, monthly income, comorbidities, and psychiatric problems. The injury level did not affect the incidence of alcohol dependence after SCI.

## 1. Introduction

The spinal cord connects the motor and perception channels between the brain and the limbs. Damage to the spinal cord hinders the transmission of information from the injured area, the higher the site of SCI, the more dysfunction remains [1]. Concentration of microtubule associated protein tau in urine and saliva was a potential biomarker of traumatic brain injury [2]. Binge drinking and alcohol dependence are raised over the general population at the time of SCI [3]. A high prevalence of alcohol use has been documented among post-injury SCI persons, limited research exists on alcohol use and on longitudinal studies [4]. SCI is a devastating disorder. The most common causes of SCI are car accidents, falls, violence, diving incidents, or medical/surgical complications [5]. Survivors usually suffer from motor impairment, bladder or bowel dysfunction, autonomic dysfunction, infection, pulmonary complications, pain, and psychosocial issues [6,7,8,9,10,11,12,13]. Patients also encounter depression, anxiety, and even cognitive impairment [14,15,16], which have a negative impact on mental health and quality of life [17]. Some patients use cannabis, while others seek alcohol, to relieve pain and psychological problems [18,19,20,21]. One study revealed the fact that alcohol abuse is quite common in SCI groups [22]. Another study pointed that depression impairs the quality of life among persons with alcohol dependence [23]. A study indicated 19.3% were heavy drinkers, and 29.4% were moderate drinkers after SCI [20]. The ω-3 fatty acids was a therapeutic treatment for SCI [24]. Binge drinking was associated with SCI severity [25]. It has been found that alcohol abuse among SCI results in a series of problems, such as pain, lower life satisfaction [26], and reduced life expectancy [27]. Patients tend to be more depressed and might experience more stress in their daily life compared with non-alcohol abusers [28]. Alcohol abuse has been recognized as hindering SCI patients from rehabilitation, thus leading to longer length of days and poorer prognosis. Therefore, the importance of identifying the possible risk factors of alcohol dependence after SCI cannot be overemphasized.

Saunders et al. [20] found that personality and socioeconomic status are associated with heavy drinking in SCI patients. Male gender is also an important risk factor of heavy drinking; however, age is negatively associated with this issue. Tate et al. [26] also indicated that younger, single, male, and less educated patients are at risk of being alcohol abusers. In addition, alcohol abuse post-SCI is usually a continuation of problem drinking that occurred before SCI [29]. However, there are still no studies associating the disease severity and comorbidities with alcohol dependence in SCI patients. Therefore, we conducted this study to understand the associations between alcohol dependence and related risk factors in SCI patients.

## 2. Materials and Methods

### 2.1. Study Design and Data Source

The study protocol was approved by the Institutional Review Board/Chang Gung Memorial Hospital (IRB/CGMH) (approval number 104-5772B). Data were obtained from the Longitudinal Health Insurance Database 2005 (LHID2005) of the NHIRD. The LHID consists of medical claims of one million beneficiaries randomly selected from all insured Taiwanese in the year 2005, with age and sex distributions being almost identical to the general population. We retrieved the longitudinal data of inpatient claims, outpatient claims, and registry claims data from 2000 to 2009.

### 2.2. Participants

Subjects with a diagnosis (International Classification of Diseases, 9th revision, Clinical Modification (ICD-9-CM) codes 806, 952) available from the inpatient or outpatient claims data were enrolled during the period 2000 to 2009. The level of injury could be categorized into four groups, including cervical SCI (ICD-9-CM 806.00–806.19, 952.00–952.09), thoracic SCI (ICD-9-CM 806.20–806.39, 952.10–952.19), lumbar SCI (ICD-9-CM 806.4, 806.5, 952.2), and other SCI (ICD-9-CM 806.60–806.9, 952.3–952.9, 806, 952). The exclusion criteria were subjects with a diagnosis of SCI from before 2000, those aged less than 20 years old, or those with incomplete or missing data from the database. Propensity score matching was used to select the control group from the claims data for comparison with the cohort group. Propensity score matching of the study cohort was performed according to the propensity scores based on age, sex, insurance level (representative of monthly income), number of visits to a psychiatric outpatient department (OPD), and Deyo’s Charlson comorbidity index (Deyo’s CCI) [30].

### 2.3. Study End Points

The first outpatient or inpatient visit for the SCI diagnosis during the study period was considered as the index date. The SCI unmatched groups were significantly diverse in age, gender, and comorbidity. Patients without SCI were used the propensity score matched approach at the start of the procedure. The matched SCI groups using the 1:1 propensity score to create groups matched at age, gender, comorbidity, and monthly income to reduce selection bias. The follow-up end points of each case were set at the time points when the subjects were newly diagnosed as having alcoholism. We divided the two cohorts into the two groups: patients with SCI and those without SCI. In both cohorts, alcoholism diagnoses were observed only following the index date during the follow up period. The study endpoint was the subsequent development of with alcoholism were defined as having an inpatient or outpatient diagnosis of alcohol withdrawal delirium (ICD-9-CM 291.0), alcohol amnestic syndrome (ICD-9-CM 291.1), alcoholic dementia (ICD-9-CM 291.2), alcohol withdrawal hallucinosis (ICD-9-CM 291.3), idiosyncratic alcohol intoxication (ICD-9-CM 291.4), alcoholic jealousy (ICD-9-CM 291.5), alcoholic psychoses (ICD-9-CM 291.9), alcoholic drunkenness (ICD-9-CM 303.00), alcohol dependence syndrome (alcohol intoxication) (ICD-9-CM 303.90), alcoholic gastritis (ICD-9-CM 535.30), alcoholic liver cirrhosis (ICD-9-CM 571.2), or alcoholic liver disease (ICD-9-CM 571.3) from the index date until the end of Subjects (Figure 1).

### 2.4. Statistical Analysis

The baseline characteristics between the SCI and non-SCI groups were compared by the independent t test or chi-square test. The numbers of SCI subjects with alcohol dependence syndrome were analyzed according to level of SCI, age, sex, insurance level, Deyo’s CCI, and number of visits to a psychiatric OPD, and the corresponding incidence rates were also reported. The cumulative incidence rate of alcohol dependence syndrome between the SCI and non-SCI groups were compared by the log-rank test. We further analyzed the hazard ratios (HR) of alcohol dependence syndrome according to the above covariates by the Cox proportional hazard model. Statistical significance was set at a level of *p*-value < 0.05. All statistics were analyzed using SAS version 9.4 (SAS Institute Inc., Cary, NC, USA).

## 3. Results

A total of 5670 subjects in the SCI group and 705,937 subjects in the non-SCI group were retrieved from the database. The baseline characteristics in the entire unmatched cohort varied between the two groups. Subjects in the SCI group were more likely to be older, less males, have a lower insurance level, higher Deyo’s CCI, and have higher ratios of multiple visits to a psychiatric OPD. After 1:1 propensity score matching, a total of 27,719 subjects, including 5639 from the SCI group and 22,080 from the non-SCI group, were included in the analysis. All baseline characteristics were similar between the SCI and non-SCI groups (Table 1).

In the SCI group, 55 subjects were diagnosed as having alcohol dependence syndrome, while the other 5584 subjects were not. The distribution of level of SCI and Deyo’s CCI were similar between the alcohol dependence syndrome and non-alcohol dependence syndrome groups. However, those with alcohol dependence syndrome were younger (43.7 ± 12.6 vs. 54.1 ± 18.4 years old). The ratio of males was higher in the subjects with alcohol dependence syndrome (87.3% vs. 47.2%). The proportion of subjects with a low insurance level (≤New Taiwan Dollar NTD 15,840 per year) was also greater in the subjects with alcohol dependence syndrome (47.3% vs. 25.6%). There were more chances of the subjects with alcohol dependence syndrome to visit a psychiatric OPD multiple times (≥2 times) (23.6% vs. 12.8%). The above findings are shown in Table 2.

The cumulative incidence rate of alcohol dependence syndrome between the SCI and non-SCI groups were significantly different (log-rank *p* = 0.0247) (Figure 2). The possible risk factors associated with alcohol dependence syndrome in the SCI patients were analyzed and are shown in Table 3. A significantly greater hazard of alcohol dependence syndrome was noted in the SCI group compared to the non-SCI group (adjusted HR: 1.39, 95% CI: 1.03~1.86, *p* = 0.0305). The level of SCI was not associated with the incidence of alcohol dependence syndrome. A higher incidence of alcohol dependence syndrome was also related to male patients (adjusted HR: 5.51, 95% CI: 4.13~7.35, *p* < 0.0001, vs. female), a lower insurance level (adjusted HR: 2.42, 95% CI: 1.75~3.34, *p* < 0.0001, for NTD 15,840-25,000 per year vs. >NTD 25,000 per year; adjusted HR: 2.58, 95% CI: 1.83~3.62, *p* < 0.0001, for <NTD 15,840 per year vs. >NTD 25,000 per year), higher Deyo’s CCI (adjusted HR: 1.77, 95% CI: 1.33~2.35, *p* < 0.0001, for Deyo’s CCI = 1 vs. Deyo’s CCI = 0; adjusted HR: 1.74, 95% CI: 1.29~2.36, *p* = 0.0003, for Deyo’s CCI ≥ 2 vs. Deyo’s CCI = 0), and psychiatric OPD visits (adjusted HR: 2.19, 95% CI: 1.45~3.29, *p* = 0.0002, for visiting once vs. never; adjusted HR: 2.46, 95% CI: 1.88~3.21, *p* < 0.0001, for visiting over twice vs. never). Lower incidence was related to elder age (adjusted HR: 0.977, 95% CI: 0.970~0.984, *p* < 0.0001, for per year increase).

## 4. Discussion

In our study, we utilized the NHIRD to follow up 5639 SCI patients and 22,080 non-SCI patients in Taiwan over 10 years and analyzed the possible risk factors of alcohol dependence, such as disease severity, age, sex, socioeconomic status, comorbidities, and psychiatric OPD visits. After the analysis, those with alcohol dependence syndrome tended to be younger, male, have a lower insurance level, and have more ratios of multiple psychiatric OPD visits. The cumulative incidence plot showed that the SCI group had a higher cumulative incidence of alcohol dependence syndrome than the non-SCI group. The level of SCI was not related to the incidence of alcohol dependence syndrome. The incidence of alcohol dependence syndrome was positively associated with male patients, a lower insurance level, higher Deyo’s CCI, and psychiatric OPD visits, but it was negatively associated with age.

Injury level and severity could affect the incidence of alcohol dependence after SCI. Saunders et al. [20] found a lower risk for heavy drinking in patients with high-level cervical injuries (non-ambulatory). Tate et al. [26] showed that SCI patients with ASIA grade D injuries tend to engage in abusive drinking; however, no other significant differences in alcohol consumption or abuse were observed across different levels of neurologic classification. The findings of our study were similar to those of Tate et al. [26], in which the incidence of alcohol dependence was not associated with the injury level. Because no data from the ASIA impairment scale were available from the NHIRD, whether the injury severity affected the incidence of alcohol dependence could not be proved in our study.

Alcohol dependence is believed to be associated with socioeconomic status. Saunders et al. [20] indicated that SCI patients with higher incomes or education are more likely to be heavy drinkers.

Two previous studies [31,32] support the conclusions of Saunders et al. However, our study reported an increased hazard of alcohol dependence in those with a lower income. In addition, Tate et al. [26] showed that alcohol abusers tend to be less educated. What caused the different results among Saunders et al. [20], Tate et al. [26], and our study, is possibly the selection bias. The participants of Saunders et al. [20] were from a large rehabilitation hospital in the southeastern United States. However, the participants of Tate et al. [26] were 16 Model Spinal Cord Injury Systems (MSCIS) patients from across the United States, and the participants of our study were from the NHIRD. Both MSCIS and NHIRD are large population-based databases, and the selection bias could be minimized. In fact, there are still limited studies discussing patients’ financial status after SCI. One study evaluating the financial status of SCI or traumatic brain injury (TBI) patients indicated that, compared to randomly selected patients, SCI and TBI patients are more likely to receive government income assistance at the time of bankruptcy [33]. In addition, social roles and peer pressure might affect an individual’s behavior. This study found that young males tend to use alcohol to release the pressure from others after the occurrence of injury, and this phenomenon has been reported in other studies [20,26].

Co-occurring somatic and psychiatric conditions are often related to alcohol dependence in the general population [34]. Alcohol dependence has been found to be associated with major diseases such as liver cirrhosis, tuberculosis, cancer, diabetes mellitus, ischemic heart disease, stroke, or epilepsy, leading to higher mortality rates and poor health-related quality of life [35]. In addition to somatic problems, psychiatric disorders usually come with alcohol dependence. The cause–effect relationship between psychiatric disorders and alcohol dependence is difficult to define. Antisocial personalities and depressive disorders often predate the onset of alcohol dependence [36], but alcohol dependence can also result in some psychiatric complaints [37]. Therefore, previous psychiatric problems were an important factor when discussing the incidence of alcohol dependence in SCI patients. Because various psychiatric complaints and diagnoses are difficult to retrieve from a population-based database, multiple psychiatric OPD visits were regarded as psychiatric complaints in our study. In summary, our findings implied that comorbidities and psychiatric problems in SCI patients influence the incidence of alcohol dependence, and the results were similar to those in the normal population.

### Study Limitations

The present study had strength in following a large SCI cohort for a long period, but it still had some limitations. First, the diagnoses of alcohol dependence syndrome from the NHIRD were based on ICD-9-CM codes and were less accurate than those diagnosed prospectively according to clinical criteria. This limitation is quite often seen in retrospective database research. Second, additional socioeconomic status data, such as occupation and education, could not be obtained from the NHIRD. These factors are believed to be associated with the incidence of alcohol dependence. Third, the data of psychological evaluations did not exist in the large database: therefore, we could not do further research about the effects of psychological factors on alcohol dependence. Prospective studies with accurate diagnoses by physicians and additional psychological evaluations could strengthen the real-world evidence about alcohol dependence in SCI patients.

## 5. Conclusions

SCI is a devastating problem to patients on not only the physical but also the psychological level. The incidence of alcohol dependence increases after the occurrence of SCI, and it is also related to age, sex, monthly income, comorbidities, and psychiatric problems. The injury level does not affect the incidence of alcohol dependence after SCI. The reported data are essential for public health and worth our attention to adjust the clinical care for SCI patients.

## Figures and Tables

**Figure 1 jpm-12-00473-f001:**
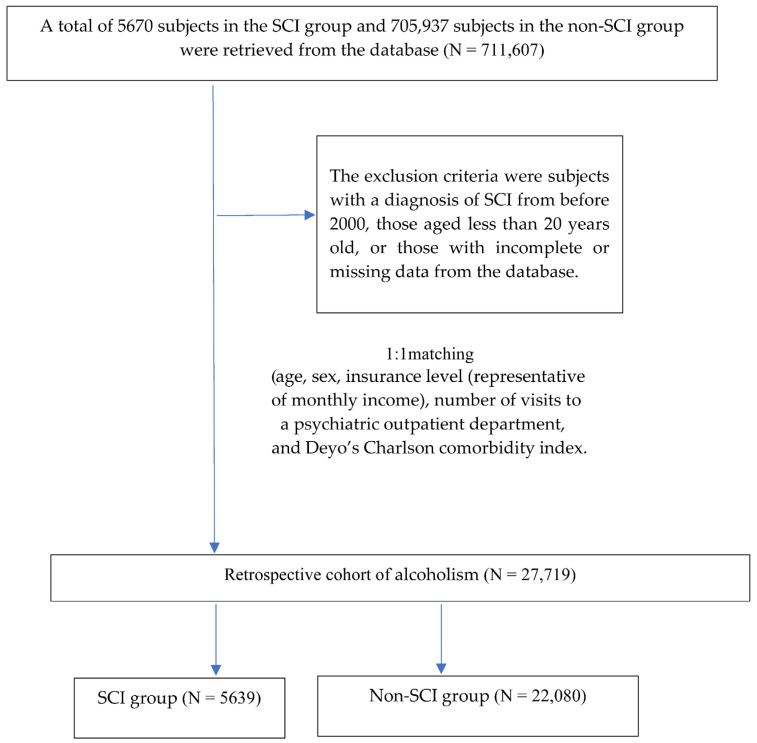
The sample selection of alcohol dependence.

**Figure 2 jpm-12-00473-f002:**
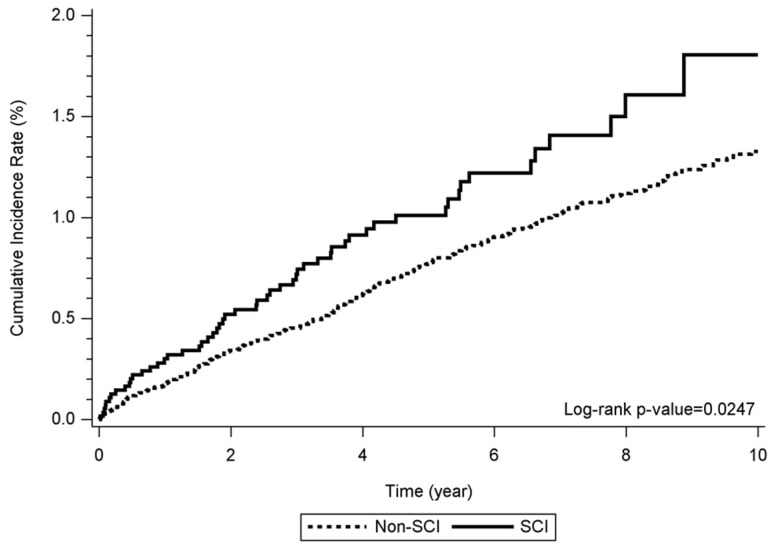
Cumulative incidence among the SCI and non-SCI groups over 10 years. Abbreviations: SCI; spinal cord injury.

**Table 1 jpm-12-00473-t001:** Baseline Characteristics of the Study Population.

Variables		Before Matching		*p*-Value	After Matching		
		SCI	Non-SCI		SCI	Non-SCI	
		(*n* = 5670)	(*n* = 705,937)		(*n* = 5639)	(*n* = 22,080)	
Age		54.2 ± 18.5	38.6 ± 15.1	<0.0001	54.0 ± 18.4	53.4 ± 18.0	0.0665
Sex	Male	2694 (47.5%)	351,273 (49.8%)	0.0008	2685 (47.6%)	10,621 (48.1%)	0.5131
	Female	2976 (52.5%)	354,664 (50.2%)		2954 (52.4%)	11,459 (51.9%)	
Insurance level (NTD $)	≤15,840	1469 (25.9%)	161,232 (22.8%)	<0.0001	1457 (25.8%)	5704 (25.8%)	0.8683
	15,840–25,000	2804 (49.5%)	307,082 (43.5%)		2789 (49.5%)	10,850 (49.1%)	
	>25,000	1397 (24.6%)	237,623 (33.7%)		1393 (24.7%)	5526 (25.0%)	
Deyo’s CCI	0	2369 (41.8%)	608,474 (86.2%)	<0.0001	2368 (42.0%)	9464 (42.9%)	0.2252
	1	1212 (21.4%)	66,371 (9.4%)		1209 (21.4%)	4814 (21.8%)	
	≥2	2089 (36.8%)	31,092 (4.4%)		2062 (36.6%)	7802 (35.3%)	
Psychiatric OPD visits	0	4684 (82.6%)	628,024 (89.0%)	<0.0001	4672 (82.9%)	18,381 (83.2%)	0.6039
	1	243 (4.3%)	21,445 (3.0%)		239 (4.2%)	874 (4.0%)	
	≥2	743 (13.1%)	56,468 (8.0%)		728 (12.9%)	2825 (12.8%)	

Abbreviations: SCI, spinal cord injury; CCI, Charlson Comorbidity Index; OPD, outpatient department; NTD, New Taiwan Dollar.

**Table 2 jpm-12-00473-t002:** Distribution of the subjects with alcohol dependence syndrome in the study cohort.

Variables	Alcohol Dependence Syndrome	No Alcohol Dependence Syndrome	*p*-Value
	(*n* = 55)	(*n* = 5584)	
Level of SCI			0.9536
Cervical	18 (32.7%)	1676 (30.0%)	
Thoracic	8 (14.5%)	778 (13.9%)	
Lumbar	13 (23.6%)	1487 (26.6%)	
Other	16 (29.1%)	1643 (29.4%)	
Age	43.7 ± 12.6	54.1 ± 18.4	<0.0001
Sex			<0.0001
Male	48 (87.3%)	2637 (47.2%)	
Female	7 (12.7%)	2947 (52.8%)	
Insurance level (NTD $)			0.0006
≤15,840	26 (47.3%)	1431 (25.6%)	
15,840–25,000	23 (41.8%)	2766 (49.5%)	
>25,000	6 (10.9%)	1387 (24.8%)	
Deyo’s CCI			0.1995
0	22 (40.0%)	2346 (42.0%)	
1	17 (30.9%)	1192 (21.3%)	
≥2	16 (29.1%)	2046 (36.6%)	
Psychiatric OPD times			0.0080
0	37 (67.3%)	4635 (83.0%)	
1	5 (9.1%)	234 (4.2%)	
≥2	13 (23.6%)	715 (12.8%)	

Abbreviations: CCI, Charlson Comorbidity Index; OPD, outpatient department; NTD, New Taiwan Dollar.

**Table 3 jpm-12-00473-t003:** Incidence of alcohol dependence syndrome according to the covariates.

All Alcohol Dependence Syndrome		Crude Model	*p*-Value	Adjusted Model	*p*-Value
Variables	Comparison	HR (95% C.I.)		HR (95% C.I.)	
SCI	Yes vs. No	1.40 (1.04, 1.88)	0.0252	1.39 (1.03, 1.86)	0.0305
Level of SCI	No SCI	1		—	—
	Cervical	1.56 (0.96, 2.52)	0.0702	—	—
	Thoracic	1.56 (0.77, 3.16)	0.2156	—	—
	Lumbar	1.21 (0.69, 2.12)	0.4995	—	—
	other	1.35 (0.81, 2.24)	0.2492	—	—
Age	Per 1 year increase	0.982 (0.975, 0.988)	<0.0001	0.977 (0.970, 0.984)	<0.0001
Sex	Female	1		1	
	Male	5.15 (3.88, 6.85)	<0.0001	5.51 (4.13, 7.35)	<0.0001
Insurance level (NTD $)	<15,840	2.39 (1.71, 3.35)	<0.0001	2.58 (1.83, 3.62)	<0.0001
	15,840–25,000	1.74 (1.27, 2.40)	0.0007	2.42 (1.75, 3.34)	<0.0001
	>25,000	1		1	
Deyo’s CCI	0	1		1	
	1	1.40 (1.06, 1.84)	0.0180	1.77 (1.33, 2.35)	<0.0001
	≥2	1.16 (0.90, 1.50)	0.2564	1.74 (1.29, 2.36)	0.0003
Psychiatric OPD visits	0	1		1	
	1	2.43 (1.62, 3.65)	<0.0001	2.19 (1.45, 3.29)	0.0002
	≥2	2.24 (1.73, 2.90)	<0.0001	2.46 (1.88, 3.21)	<0.0001

Abbreviations: SCI, spinal cord injury; CCI, Charlson Comorbidity Index; OPD, outpatient department; HR, hazard ratio; C.I., confidence interval; NTD, New Taiwan Dollar.

## Data Availability

The original data are available upon reasonable request to the corresponding author.

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
