# Peer review of "Association between Spinal Cord Injury and Alcohol Dependence: A Population-Based Retrospective Cohort Study"

_jpm, 2022, doi:10.3390/jpm12030473_

Round 1
Reviewer 1 Report
The original article by Chuang et al. "Association Between Spinal Cord Injury and Alcohol Dependence: A Population-Based Retrospective Cohort Study" covers a potentially interesting and emerging topic related to the SCI and alcohol abuse. In this sense, this remains to be potentially interesting for the JPC readers. I regard the main point of this paper as highly attractive as well as the results are clearly presented.The manuscript posess hight citation potential.The text does not contain any major errors, therefore I have some minor comments and recommendations:
- There is a need to provide slightly more expanded introduction regarding SCI pathogenesis. Could alcohol influence this regarding pathophysiology of "secondary injury"?
- The figure/schema summarizing and clarifying the study design should be added.
- In some places the use of English could be improved on.
- Following references should be added and properly cited within the main text:
- Olczak M, Poniatowski ŁA, Niderla-Bielińska J, Kwiatkowska M, Chutorański D, Tarka S, Wierzba-Bobrowicz T. Concentration of microtubule associated protein tau (MAPT) in urine and saliva as a potential biomarker of traumatic brain injury in relationship with blood-brain barrier disruption in postmortem examination. Forensic Sci Int. 2019 Aug;301:28-36. doi: 10.1016/j.forsciint.2019.05.010.
- Davis JF, Cao Y, Krause JS. Changes in alcohol use after the onset of spinal cord injury. J Spinal Cord Med. 2018 Mar;41(2):230-237. doi: 10.1080/10790268.2017.1319996.
- Wojdasiewicz P, Poniatowski ŁA, Turczyn P, Frasuńska J, Paradowska-Gorycka A, Tarnacka B. Significance of Omega-3 Fatty Acids in the Prophylaxis and Treatment after Spinal Cord Injury in Rodent Models. Mediators Inflamm. 2020 Jul 29;2020:3164260. doi: 10.1155/2020/3164260.
- Saunders LL, Krause JS, Carpenter MJ, Saladin M. Risk behaviors related to cigarette smoking among persons with spinal cord injury. Nicotine Tob Res. 2014 Feb;16(2):224-30. doi: 10.1093/ntr/ntt153.
Completing all this gaps will have an impact on the understanding the aim of the study and, from my point of view, is absolutely necessary.
Author Response
Reviewer 1 comments:
|
Reviewer 1 |
Reply |
|
There is a need to provide slightly more expanded introduction regarding SCI pathogenesis. Could alcohol influence this regarding pathophysiology of "secondary injury"? |
The spinal cord connects the motor and perception channels between the brain and the limbs, damage to the spinal cord hinders the transmission of information from the injured area, more high site of spinal cord injury, the more dysfunction remains. Neuroconvulsions, and respiratory restrained occur most often in the early stages. Loss of sensory, motor, and autonomic function resulting in flaccid paralysis of limbs, retention of urine, irregular contractions of the colon, slowed peristalsis in three months. High prevalence of alcohol use has been documented among post-injury SCI persons, limited research exists on alcohol use and on longitudinal studies |
|
The figure/schema summarizing and clarifying the study design should be added. |
As revised Figure 1 |
|
In some places the use of English could be improved on. |
We have revised it. |
|
Following references should be added and properly cited within the main text: · Olczak M, Poniatowski ŁA, Niderla-Bielińska J, Kwiatkowska M, Chutorański D, Tarka S, Wierzba-Bobrowicz T. Concentration of microtubule associated protein tau (MAPT) in urine and saliva as a potential biomarker of traumatic brain injury in relationship with blood-brain barrier disruption in postmortem examination. Forensic Sci Int. 2019 Aug;301:28-36. doi: 10.1016/j.forsciint.2019.05.010. · Davis JF, Cao Y, Krause JS. Changes in alcohol use after the onset of spinal cord injury. J Spinal Cord Med. 2018 Mar;41(2):230-237. doi: 10.1080/10790268.2017.1319996. · Wojdasiewicz P, Poniatowski ŁA, Turczyn P, Frasuńska J, Paradowska-Gorycka A, Tarnacka B. Significance of Omega-3 Fatty Acids in the Prophylaxis and Treatment after Spinal Cord Injury in Rodent Models. Mediators Inflamm. 2020 Jul 29;2020:3164260. doi: 10.1155/2020/3164260. · Saunders LL, Krause JS, Carpenter MJ, Saladin M. Risk behaviors related to cigarette smoking among persons with spinal cord injury. Nicotine Tob Res. 2014 Feb;16(2):224-30. doi: 10.1093/ntr/ntt153. |
Following references were added and properly cited within the main text. .
|
|
Completing all this gaps will have an impact on the understanding the aim of the study and, from my point of view, is absolutely necessary. |
Thank you for your suggestions and comments. |

Reviewer 2 Report
The paper takes up a very important and rarely described problem of alcohol dependence in patients with spinal cord injury. The paper written in clear language, easy to read.
In my opinion, a major limitation of the study was the fact that the diagnosis of alcohol addiction was based on the ICD-9 classification and not on clinical criteria. Another equally important limitation of the study is the fact that the ASIA scale of people with SCI was not analyzed.
It has been shown that the level of SCI was not associated with the incidence of alcohol dependence syndrome but an ASIA analysis would, however, show whether the severity of the neurological deficits contributes to alcohol addiction.
In the introduction, the authors should write more information on the factors contributing to alcohol dependence, such as reduced quality of life, depression.
In conclusion, there is no clinical message on what to do to avoid alcoholism in this group of patients.
Author Response
Reviewers 2 comments:
|
Reviewer 2 |
Reply |
|
The paper takes up a very important and rarely described problem of alcohol dependence in patients with spinal cord injury. The paper written in clear language, easy to read. |
Thank you for your comments. |
|
In my opinion, a major limitation of the study was the fact that the diagnosis of alcohol addiction was based on the ICD-9 classification and not on clinical criteria. Another equally important limitation of the study is the fact that the ASIA scale of people with SCI was not analyzed. |
Because of data limitations, limited as well some interesting insight such as the ASIA scale of people with SCI was not analyzed. But the article proposes the use of real world data from currently existing database to determine the association of development of alcohol abuse and spinal cord injury, which can influence clinical practice. |
|
It has been shown that the level of SCI was not associated with the incidence of alcohol dependence syndrome but an ASIA analysis would, however, show whether the severity of the neurological deficits contributes to alcohol addiction. |
The fact that all the ICD-9 codes for alcohol abuse reflect a heavy drinking, therefore missing lessen alcoholism types. Therefore, it would probably find more association |
|
In the introduction, the authors should write more information on the factors contributing to alcohol dependence, such as reduced quality of life, depression. |
Another study pointed that depression impairs quality of life among persons with alcohol dependence. 19 According to Saunders et al. 16alcohol abuse after SCI groups 19.3% were heavy drinkers, 29.4% moderate drinkers |
|
In conclusion, there is no clinical message on what to do to avoid alcoholism in this group of patients. |
this study offers useful information to understand the alcohol dependence of persons with SCI. An Early mental health interventions might include providing guidance for counseling of individual persons with SCI, and facilitating the development of mental health support groups. |
